# dsCleaner: A Python Library to Clean, Preprocess and Convert Non-Intrusive Load Monitoring Datasets

**Manuel Pereira** [1,2], **Nuno Velosa** [1,2] **and Lucas Pereira** [1,3]*

1    ITI, LARSyS, 9020-105 Funchal, Portugal
2    Ciências Exatas e Engenharia, Universidade da Madeira, 9020-105 Funchal, Portugal
3    Ténico Lisboa, Universidade de Lisboa, 1049-001 Lisbon, Portugal
*    Correspondence: lucas.pereira@m-iti.org

**Abstract:**  Datasets play a vital role in data science and machine learning research as they serve as the basis for the development, evaluation, and benchmark of new algorithms. Non-Intrusive Load Monitoring is one of the fields that has been benefiting from the recent increase in the number of publicly available datasets. However, there is a lack of consensus concerning how dataset should be made available to the community, thus resulting in considerable structural differences between the publicly available datasets. This technical note presents the DSCleaner, a Python library to clean, preprocess, and convert time series datasets to a standard file format. Two application examples using real-world datasets are also presented to show the technical validity of the proposed library.

**Keywords:** datasets; NILM; library; python; cleaning; preprocessing; conversion

## 1. Introduction

Datasets play a vital role in data science and machine learning research as they serve as the basis for the development, evaluation, and benchmark of new algorithms. Smart-Grids are among the research fields that have seen an increase in the number of available public datasets. An example is the subfield of Non-Intrusive Load Monitoring (NILM) [1], which according to [2], has now more than 20 publicly available datasets.

NILM is the process of estimating the energy consumption of individual appliances from electric power measurements taken at a limited number of locations in the electrical distribution of a building. As such, suitable NILM datasets consist of time-series measurements from the whole-house demand (taken at the mains), and of the individual loads (i.e., ground-truth data). The individual load consumption is obtained by measuring each load at the plug-level, or by measuring the individual circuit to which the loads are connected [2].

Yet, despite decades of research ([3–5]) and recent efforts towards improving the performance evaluation of the proposed methods, either by creating public datasets (e.g., REDD [6], BLUED [7], UK-DALE [8], AMDds, [9], and REFIT [10]), studying performance metrics (e.g., [11–13]), or developing tools to run evaluations across different datasets [14], the ability to perform formal evaluations of this technology is still an open research question.

We argue that one of the main reasons behind the limited impact of public datasets is the absence of a practical and easy to use mechanisms to access and manipulate the contents of the NILM datasets. Heterogeneity in NILM datasets comes in different flavors, each of which posing different challenges. For example:

- **Different File Formats:** datasets come in several file formats, text files are the more prominent (e.g., BLUED [7]. Likewise, it is also possible to find datasets that use formats such as FLAC[1] [8], WAVE[2] [15], and relational databases [16]. Consequently, before conducting any performance evaluation researchers must first understand how the data is formatted and change their algorithms accordingly, including the calculation of performance metrics.

- **Different Sampling Rates and Missing Data:** datasets come in a variety of sampling rates from 1/60 Hz to several kHz or even MHz in some extreme cases [2]. Furthermore, due to the complexity of the hardware installations to collect such datasets, it is very common to have considerable periods of missing data [14,17]. This often requires the adaptation of algorithms to cope with a different sampling rate, and with missing data since some algorithms may assume the existence of continuous data.

- **Number of Files and Folder Structure:** datasets are made available in several files and different folder structures. For example, when uncompressed, BLUED takes about 320 GB of disk space, which include 6496 files of distributed across 16 folders. This requires the development of additional code just for data loading, and to cope with different compression strategies. Furthermore, since existing algorithms are likely to support a specific type of data input (e.g., a single file containing all the data, or hourly files), it may be necessary to adapt the algorithms to the new folder structure or vice-versa.

Against this background, reducing the heterogeneity among NILM datasets has been the subject of attention from a small number of researchers.

In Kelly and Knottenbelt [18], the authors proposed a hierarchical metadata schema for energy disaggregation (NILM Metadata). The proposed schema is divided into two main parts: (1) central metadata with general information about how appliances are represented in NILM metadata; and (2) a schema describing meters, buildings, and datasets.

In Batra et al. [14], the authors proposed NILMTK, an open-source toolkit aimed at providing a unified approach to explore existing datasets and evaluate NILM algorithms. In NILMTK, the datasets are represented using the NILMTK data format (NILMTK-DF), which is based on the hierarchical data format (HDF5). Noticeably, this toolkit relies on the schema defined by the NILM metadata project to provide a common interface with the different datasets. Among the features for dataset analysis are: (1) loaders and converters; (2) data diagnosis; (3) statistical analysis; and (4) preprocessing.

In Pereira [15], the authors proposed the energy monitoring and disaggregation data format (EMD-DF), which is a common data model and file format to support the creation and manipulation of load disaggregation datasets. The EMD-DF data model defines three main data entities that should be present in a dataset for load disaggregation: (1) consumption data; (2) ground-truth data; and (3) data annotations. In the same work, the authors also present an implementation of the data model by extending the wave-form audio format (WAVE). One of the main features of the proposed implementation is the possibility of embedding the data annotations along with the aggregated and ground-truth consumption data. Noticeably, the authors also proposed including the NILM metadata schema as part of the data annotations.

The size of the datasets, in particular for sampling rates in the orders of the several kHz, was also the subject of attention from a group of researchers. In Kriechbaumer et al. [19], the authors conducted a comprehensive benchmark of 365 data representation formats against five high-frequency datasets (12 kHz $\geq$ frequency $\leq$ 250 kHz). Ultimately, this paper shows that by selecting a suitable file format and data transformation, it is possible to save up to 73% in disk space. Furthermore, this work also advocates in favor of using audio-based file formats due to the similarities with electricity waveforms.

---

[1]　FLAC, http://fileformats.archiveteam.org/wiki/FLAC
[2]　WAVE, http://fileformats.archiveteam.org/wiki/WAV

This technical note contributes to the ongoing body of work by proposing *dsCleaner*, a Python library to clean, preprocess, and convert NILM datasets to a standard file format. The remaining of the paper is organized as follows: first, Section 2 provides a thorough description of the *dsCleaner*. Then, Section 3, presents two application examples. Section 4 provides an overview of the online resources before the paper concludes in Section 5 with an outline of future work possibilities towards better interoperability across datasets.

## 2. dsCleaner Library

The *dsCleaner* library aims at mitigating the abovementioned limitations, namely, different file formats, different samples rates and missing data, and different number of files and folder structure.

The main features of the current version of *dsCleaner* are: (1) a set of functions to load datasets from CSV and media files such as WAVE and FLAC; (2) a set of functions for data cleaning, trimming and re-sampling; and (3) a set of functions to convert and merge the resulting datasets to the WAVE and Sony Wave64[3] data formats.

### 2.1. Data Processing Workflow

A high-level overview of the data processing workflow is given in Figure 1. Additional details of each step are provided in Section 2.2.

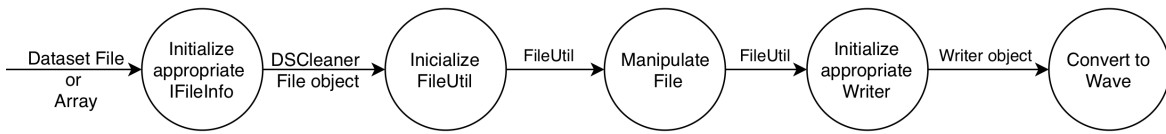

**Figure 1.** Dataflow diagram of the dsCleaner workflow.

The process starts by loading an existing dataset, that depending on the underlying file format, will also initialize the appropriate IFileInfo specialization. This is followed by the initialization of the FileUtil class that is responsible for all the data preprocessing functions. Once all the data preprocessing is performed, the appropriate FileWriter is instantiated, such that it is possible to write the dataset in the desired file format.

### 2.2. Library Overview

The UML class diagram of *dsCleaner* is given in Figure 2. The library is composed by six (6) base classes: *IFileInfo*, *FileInfo*, *CSVFileInfo*, *Merger*, *FileWriter* and *FileUtil*, all of them can be used with *with* statements, which are particularly useful since we are mostly working with real files, are prone to IO Exceptions. Each class is detailed in the following subsections.

---

3    Sony Wave64, http://fileformats.archiveteam.org/wiki/Sony_Wave64

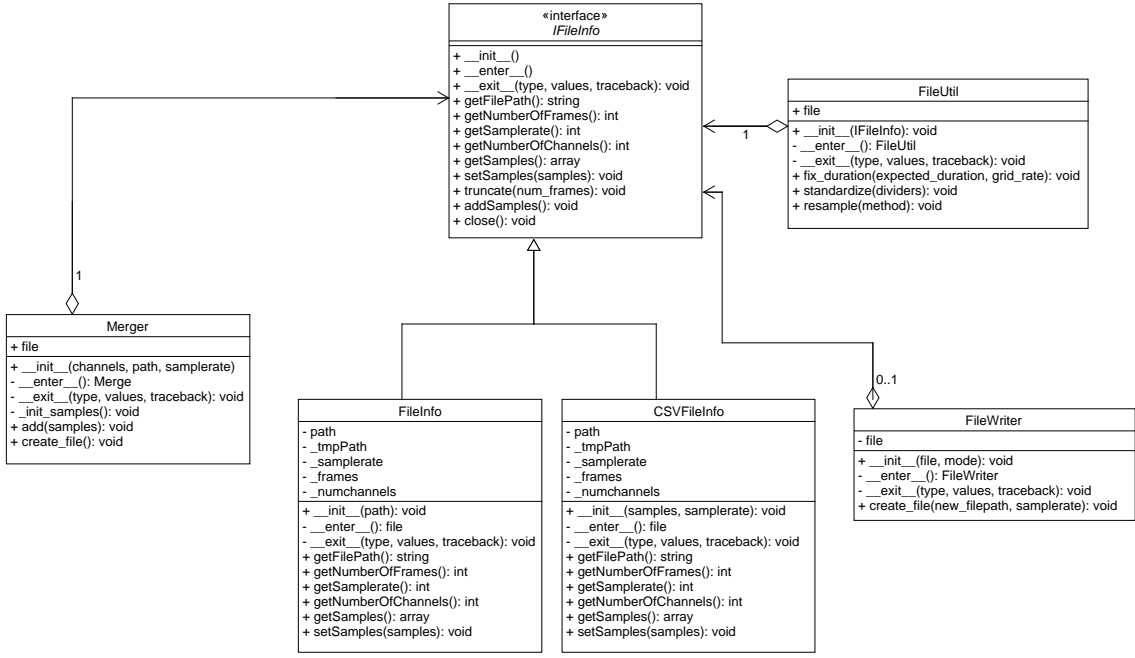

**Figure 2.** dsCleaner's Class Diagram.

### 2.2.1. IFileInfo

IFileInfo is an interface and defines all the operations that are allowed on an existing dataset. IFileInfo also adds an abstraction layer to the FileUtil, FileWriter and Merger classes. In the current version of *dsCleaner* there are two implementations of this interface, namely the FileInfo and CSVFileInfo, that were developed to support audio and text files, respectively. To support additional file formats, this class should be extended.

### 2.2.2. FileInfo

FileInfo is a specialization of IFileInfo and was implemented to support datasets that are stored using audio file formats (e.g., UK-DALE and SustDataED). It relies heavily upon PySoundFile, which is a Python implementation of the very well known C library libsndfile[4]. To use FileInfo, the only information needed is a path to a sound file. The metadata (sample rate, etc.) is completed automatically based on the file header. In Section 3.1, an example is provided using the UK-DALE dataset.

### 2.2.3. CSVFileInfo

CSVFileInfo is another specialization of IFileInfo and was implemented to support datasets stored in text formats (e.g., CSV and Tab-Separated Values). This class takes as arguments, a multi-dimensional array of samples, and the sample rate. Note that before using this class, it is necessary to convert the content of the original text files to the array format. This process is external to *dsCleaner*. In Section 3.2, an example is provided using the BLUED dataset.

### 2.2.4. FileUtil

FileUtil is the class that is responsible for performing data cleaning and preprocessing tasks. It tackles the issues of having different sampling rates and missing data. FileUtil currently supports

---

4    libsndfile, https://github.com/erikd/libsndfile

three tasks: (1) time correction; (2) re-sampling; and (3) normalization. The next subsections provide additional details for each feature.

**Time Correction**

Due to time synchronization issues that may occur in the data acquisition hardware, the files on a dataset may have more or fewer samples than the expected. This is the case of the voltage and current files in UK-DALE that despite being one-hour long files sampled at 16 kHz, often have less than 57,600,000 samples (i.e., 16k samples $\times$ 60 s$\times$ 60 min). In contrast, SustDataED [20], which is also composed on one-hour long files sampled at 12.8 kHz, often has files with more than the expected 46,080,000 samples, as shown in Figure 3.

| Name ^ | Type | Size | Length |
|---|---|---|---|
| 1475708700932_1475712300932_3600000_0_1.wav | WAV File | 180.051 KB | 01:00:01 |
| 1475712301669_1475715900932_3599263_737_2.wav | WAV File | 180.001 KB | 01:00:00 |
| 1475715901430_1475719900932_3599502_498_3.wav | WAV File | 180.001 KB | 01:00:00 |
| 1475719501174_1475723100932_3599758_242_4.wav | WAV File | 180.051 KB | 01:00:01 |
| 1475723101922_1475726700932_3599010_990_5.wav | WAV File | 180.001 KB | 01:00:00 |
| 1475726701667_1475730300932_3599265_735_6.wav | WAV File | 180.001 KB | 01:00:00 |
| 1475730301408_1475733900932_3599524_476_7.wav | WAV File | 180.001 KB | 01:00:00 |
| 1475733901168_1475737500932_3599764_236_8.wav | WAV File | 180.051 KB | 01:00:01 |

**Figure 3.** Example of files with extra samples.

Two different methods are presented to fix these issues:

1.  When there are more samples than expected, the files are truncated to the expected length. Note that the truncate method is defined in the IFileInfo interface. As such, its implementation is required in all the realizations of this interface.

2.  When there are fewer samples than expected, the first solution is to replicate the samples of the last full cycle until the end of the file. Note that this is only possible when there is no change in power during the last full cycle. Otherwise, this change would propagate until the end of the file. Consequently, the alternative is to replicate the first full cycle of the next file in the dataset. Figure 4 shown an example of the first solution.

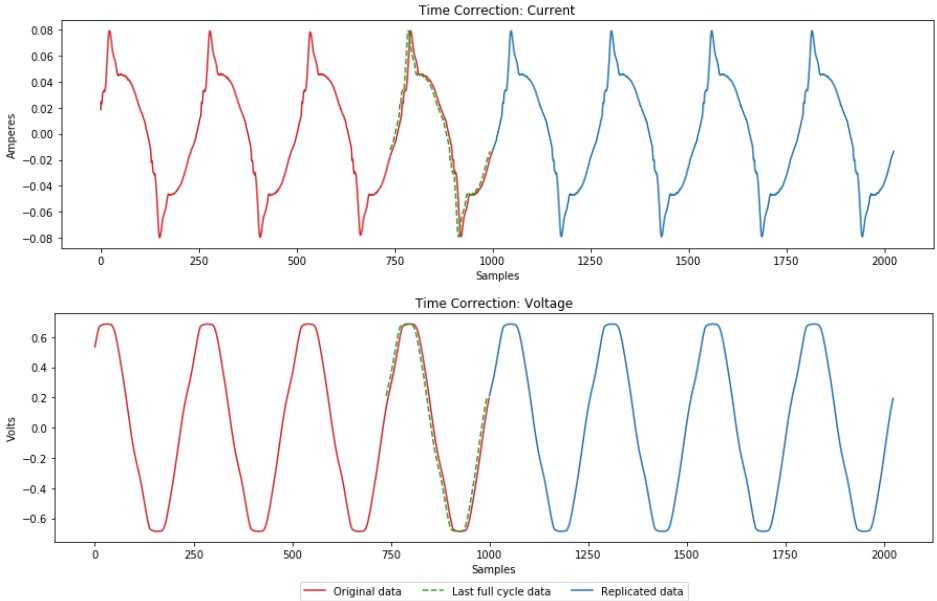

**Figure 4.** Example of time correction by sample replication: current (**top**), voltage (**bottom**).

**Re-sampling**

Since datasets come in a variety of sampling rates, it is important to have a tool to quickly re-sample the data. Likewise, by using re-sampling, it is also possible to fix problems with missing data and reduce the overall size of a dataset, more specifically via down-sampling. In dsCleaner, the re-sampling operations are performed using the Librosa[5] library. By default, the method used is fast kaiser algorithm, but it is possible to set other methods programmatically. Figure 5 shown an example of the re-sampling process in the UK-DALE dataset.

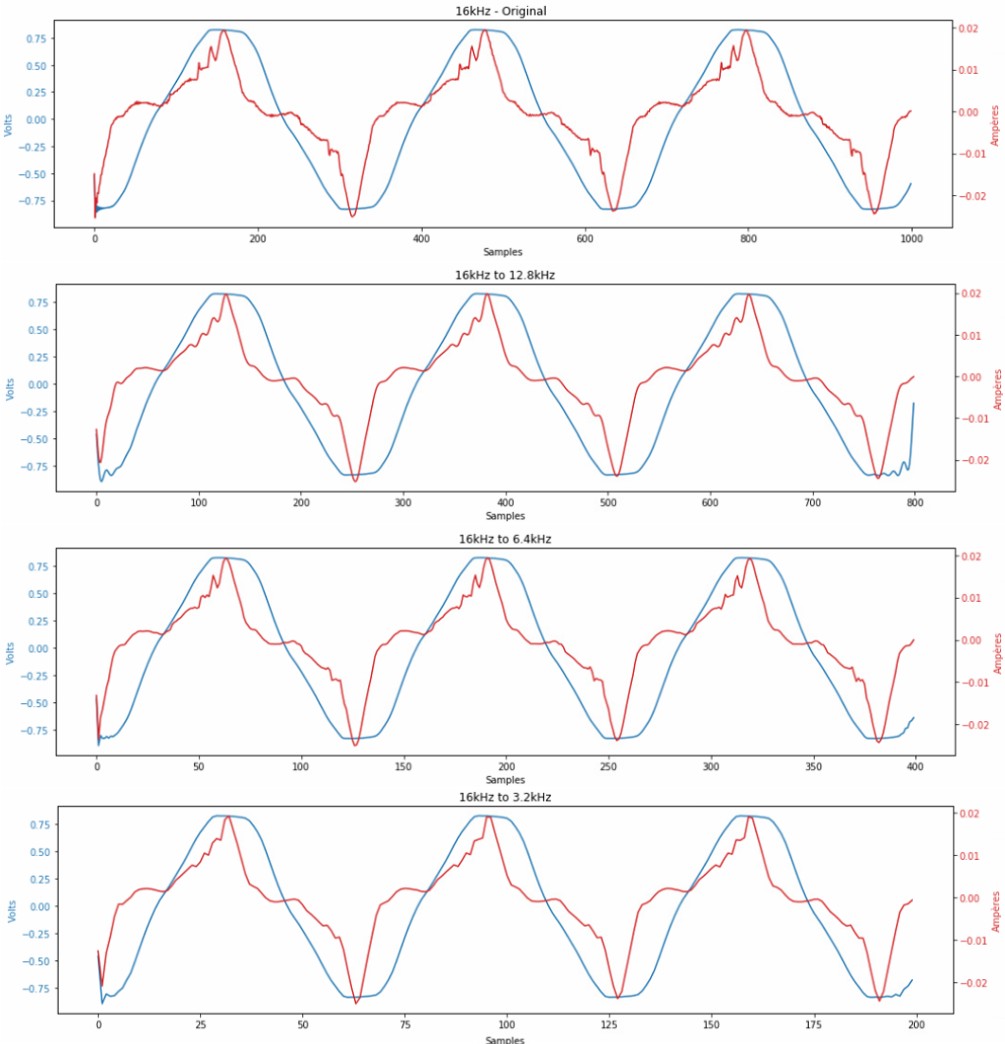

**Figure 5.** Resampling example: original signal at 16 kHz vs. resampled signal at (12.8, 6.4 , 3.2) kHz.

**Normalization**

When converting a CSV (or other text formats) dataset to audio-bases formats, the values are not normalized, i.e., not between $-1$ and 1. Thus, they cannot be correctly represented (see Figure 6). In *dsCleaner*, the standardization step is very straightforward and consists of dividing all the samples on a channel (or column) by a value greater or equal to the maximum value. This value can be defined manually, or default to the maximum value in the sequence.

---

5     Librosa, https://librosa.github.io/librosa/

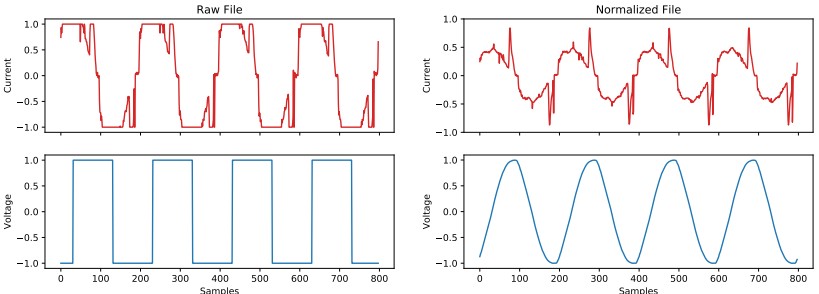

**Figure 6.** Normalization example: raw file (**left**), normalized file (**right**).

### 2.2.5. FileWriter

FileWriter is responsible for converting datasets to a different file format. It tackles the issue with different file formats by allowing to write datasets to any audio-based file format. It takes as arguments a IFileInfo specialization, and a path indicating where to save the newly created file(s). It is important to remark that the data will be converted based on the extension given in the path.

The present version of *dsCleaner* allows the conversion of datasets to either the 32-bits WAVE or the 64-bit SonyWave audio formats. WAVE files are limited to a maximum of 4 GB, due to the 32-bit headers. This size is enough to represent about 16 h of the three 16-bit channels sampled at 12 kHz (i.e., two currents phases and one voltage phase) in a single file. In contrast, the maximum size of Sony Wave64 files is of about 16 exabytes, which is equivalent to roughly 21 years of three 16-bits channels sampled at 4.3 GHz. Furthermore, since Sony Wave64 is only a 64-bits extension of the WAVE format, the backward compatibility of datasets is guaranteed by default.

Besides this, there are several other reasons to support these two audio-based formats for storing NILM datasets:

- Both formats store the data in separate channels, therefore allowing several different measurements on the same file while still providing a clear way of separating them.
- The resulting files are optimized to have very little overhead. Additionally, since the sampling rate is a fixed value, only the initial timestamp is required to obtain the timestamp of the remaining samples. Hence reducing the file size even further.
- Both are uncompressed lossless formats, i.e., all the original values of the data are kept untouched. Furthermore, it is possible to further compress these formats using lossless compression (e.g., WavePack[6].
- Since these are standard formats it is possible to find implementations in most of the existing programming languages. As such, researchers can focus on the actual NILM problem and not in finding ways to interface the datasets.

### 2.2.6. Merger

The Merger class enables the creation of data files with different sizes. It tackles the third problem mentioned in the introduction, number of files and folder structure, by allowing the merge of a group of files to a single file. It is important to note that this class also allows writing to different file formats like FileUtil.

A Merger instance takes as arguments, a path to a file (existing or not), the number of channels, the desired sample rate, and, optionally, a write mode either *w* for writing (truncating) or *a* for appending to the file. By default, if there is already a file in the given path, the append mode will be used. On the opposite case, a new file is created, and the content appended once the *create_file* method is invoked.

---

6    WavePack, http://www.wavpack.com/

## 3. Application Examples

This section provides two application examples of *dsCleaner*. The source-code and resulting datasets for each case are available online at https://osf.io/vraqz/.

### 3.1. UK-DALE

The UK-DALE dataset consists of power demand data from five houses in the UK. For each home, both the whole-house mains power demand and the power demand from individual loads, recorded every six seconds. For three of the five houses (houses 1, 2 and 5), the dataset also contains the whole-house voltage and current sampled at 16 kHz. The 16 kHz data are stored as a sequence of stereo FLAC files. Each FLAC file is about 200 MB and contains approximately 1 h of data. Channel A holds the voltage, and channel B the current.

This particular example considers one week of whole-house voltage and current data from house 2. The process starts by ensuring that each file has precisely 60 min worth of data. Each file is then re-sampled to 12.8 kHz to reduce the file size. The resulting hourly files are then merged into a single Sony Wave64 file comprised of 7 days worth of data, with a final size of 28 GB. Alternatively, it is also possible to merge the hourly files into daily Sony Wave64 files. Listing 1 shows the pseudo-code for performing the two alternatives using *dsCleaner*.

```
1  # pseudo-code to create one single file for the entire week
2  for dataset_file in file_list:
3      initialize FileInfo:
4          initialize FileUtil using FileInfo:
5              fix_duration(60)
6              resample(12800)
7          initialize Merger("ukdale_week2.w64"):
8              add(FileInfo)
9              create_file()
10
11 # pseudo-code to create one single file for each day
12 for dataset_file in file_list:
13     hour+=1
14     initialize FileInfo:
15         initialize FileUtil using FileInfo:
16             fix_duration(60)
17             resample(12800)
18         if hour % 23 == 0:
19             day+=1
20         initialize Merger(day+".w64"):
21             add(FileInfo)
22             create_file()
```

Listing 1: Python pseudo-code to process one week of data in the UK-DALE dataset.

### 3.2. BLUED

The BLUED dataset consists of power demand data from one house in the USA with a two-phase electric installation. The whole-house voltage (phase A) and current (phase A and B) sampled at 12 kHz is available. The voltage and current data are made available in text files of roughly 2:30 min each, grouped in 16 folders. Each file contains the time elapsed since the beginning of the dataset, Phase A current, Phase A voltage, and Phase B current. When fully uncompressed, BLUED takes about 353 GB of disk space.

With *dsCleaner*, it was possible to reduce the total size of BLUED to only 56 Gigabytes, roughly $\frac{1}{6}$ of the original uncompressed size. The number of files was also reduced to two (2) SonyWave64 files: Phase A and Phase B. The following formula is used to calculate the voltage for Phase B: *VoltageB* = *VoltageA* ∗ −1. Since the sound file keeps track of time by default, the time field was not necessary. It is also important to note that the original values were standardized to reduce the

bit-rate and to be represented correctly in the audio format. Listing 2 shows the pseudo-code for the different operations.

```
1  # pseudo-code to create one single file per phase for the entire dataset
2  initialize merger(PhaseA) as PhA:
3      initialize merger(PhaseB) as PhB:
4          for dataset_file in file_list:
5              csv = read_csv(dataset_file)
6              # voltage in phase B is assumed to be symetric to the voltage in phase A
7              csv.add_column(voltageB = csv['voltageA']*-1)
8              # when reading a CSV dataset the sample rate must be defined manually
9              initialize CSVFileInfo(csv['voltageA','currentA'], samplerate=12000):
10                 initialize FileUtil using CSVFileInfo:
11                     # voltage and current are divided by 180 and 80, respectively
12                     standardize(180,80)
13                 PhA.add(CSVFileInfo)
14                 PhA.create_file()
15             # the same process for Phase B
16             initialize CSVFileInfo(csv['voltageB','currentB'], samplerate=12000):
17                 initialize FileUtil using CSVFileInfo:
18                     standardize(180,80)
19                 PhB.add(CSVFileInfo)
20                 PhB.create_file()
```

Listing 2: Python pseudo-code to process the BLUED dataset.

## 4. Online Resources

### 4.1. Source Code and Documentation

*dsCleaner* is licensed under the MIT[7] license. As such it can be used, modified and redistributed. The source code is hosted in GitLab and can be accessed using the following URL: https://gitlab.com/ManelPereira/dscleaner. Comprehensive documentation has been built using Sphinx on ReadTheDocs and can be read on https://dscleaner.rtfd.io.

A stable version of dsCleaner is also available in PyPI (https://pypi.org/project/dscleaner/). To use it, the following pip command must be executed:

　　*»> pip install dscleaner*.

### 4.2. Source-Code and Example Datasets

The source code and the resulting example datasets that were used in Section 3 are also publicly available under the MIT license. The files can be found in the following Open Science Foundation repository https://osf.io/vraqz/

## 5. Conclusions and Future Work Directions

This technical note presented the initial version of *dsCleaner*, a free and open-source python library for cleaning, preprocessing, and converting NILM datasets. The project is currently under active development, and future versions of this library will include:

- Dataset splitting features, such that datasets with large files can be quickly divided into smaller ones;
- File writers for text file formats already being used by the NILM community, e.g., CSV;

---

[7] MIT License, https://opensource.org/licenses/MIT

- A command line application to provide the main *dsCleaner* features without the need for additional coding;
- Data compression features from the WavePack library, to further reduce the disk space taken by the resulting datasets;
- More complex application examples and additional metrics to quantify the applicability of *dsCleaner*. For example, the number of lines of code necessary to preprocess the datasets, or to adapt an existing algorithm to the new data format.

At this point, it is important to stress *dsCleaner* is just a first step towards the interoperability of NILM datasets.

In theory, full interoperability will happen when drop-in replacement across datasets is possible. In other words, software written to work with one dataset will work with other datasets within the same scope. Yet, in practice, interoperability is more broadly defined. For example, the Data Interoperability Standards Consortium (DISC[8]) defines interoperability as "addressing the ability of systems and services that create, exchange and consume data to have clear, shared expectations for the contents, context and meaning of that data".

In Colpaert [21], dataset interoperability is structured according to five layers: (1) legal; (2) technical; (3) syntactic; (4) semantic; and (5) querying. A brief explanation of each layer and how it applies to *dsCleaner*, and NILM datasets overall is provided next.

*Legal interoperability* mainly refers to the licensing of data publishing. When publishing an open dataset, which is normally the case with NILM, legal interoperabolity boils down to making sure that a license file is available, and that both humans and machines can understand which are the reuse rights. In NILM, the NILM metadata project provides an excellent mechanism to quickly add licensing information in a format that is already accepted by the community. The current version of *dsCleaner* does not consider legal interoperability; as such, it should also be considered in future interactions. For example, WAVE and W64 file formats have specific chunks that can be used for including copyright information (see chapter 4 of [22] for additional details).

*Technical interoperability* refers to how the data are maybe available and how they can be reached. Datasets that can be reached using standard protocols (e.g., FTP) and read using standard programming interfaces (e.g., CSV and Audio Files), have a technical interoperabilty of 100%. In the case of existing NILM datasets, this is already a common practice; as such, with rare exceptions (for example, in BLUED active and reactive power are made available using the Matlab data format), technical interoperability of datasets is almost total. *dsCleaner* aims at maximizing the technical interoperability by providing a common mechanism to access datasets thought the IFileInfo interface and its respective implementations.

*Syntactic interoperability* refers to the methods used to serialize the datasets. Two datasets have 100% syntactic interoperability if they use the same serialization and can be read using the same code base. In NILM, this would be the case of UK-DALE and SustDataED, which are serialized using the FLAC and WAVE audio-based formats, respectively. Still, by this definition, two datasets that are serialized using CSV would also have 100% syntactic intereoperability, despite the fact that it can be necessary to produce additional code to parse the data. In NILMTK, 100% syntactic interoperability is reached by parsing the datasets to the NILMTK-DF. As for *dsCleaner*, it also aims at providing 100% syntactic interoperability through the FileWrite class.

*Semantic interoperability* refers to the ability to read and compare the contents of the datasets quickly. For example, in NILM, semantic interoperability is essential for identifying the different monitored appliances and the available measurements. The NILM metadata project is presently the best option to increase the semantic interoperability of NILM datasets. In the current version of *dsCleaner*, semantic interoperability is not considered. Consequently, future interactions of *dsCleaner*

---

8    DISC, http://datainteroperability.org/

should aim at improving this aspect, for example, by adding support to the EMD-DF file format in the FileWriter class.

*Querying interoperability* refers to the ability to query a dataset and the dataset ability to reply to a given query. Ultimately, for 100% querying interoperability, only a single query interface should be available, and the dataset should contain enough information to reply to all the queries. In NILMTK, query interoperability was obtained indirectly by increasing syntactic interoperability. Nevertheless, querying interoperability also considers the ability of a dataset to reply to a given query. Still, the ability of datasets to reply to all the queries is limited. For example, a dataset that only contains the labeled transitions cannot reply to a query such as *"how much did appliance A consume in day D?"*. Likewise, a dataset that only contains aggregated consumption of the individual loads is not able to easily reply to a query such as *"how many times was appliance A used in day D?"*. The current version of *dsCleaner* already provides limited query interoperability through the IFileInfo interface, which offers a unified interface for load datasets. As such, future work should consider ways to further improve querying mechanisms of NILM datasets.

Against this background, future works should consider the possibility of conducting an in-depth search of the different problems for which NILM datasets can be used and the main data requirements of each problem. Having such information would not only help researchers to conduct more adequate data collection campaigns. But also, to have a clear understanding of how existing datasets can be "retrofitted" to enable greater interoperability.

Furthermore, having a well-defined set of requirements is the single most important aspect concerning defining which queries should be satisfied by a dataset (i.e., a minimum feature set), and which are the more appropriate data formats to support those requirements. For example, a minimum feature set would serve as a basis to perform a SWOT (strengths, weaknesses, opportunities, and threats) analysis of the existing data storage alternatives.

Finally, although this work focuses on NILM datasets, a fundamental future research direction would be to investigate the transferability of the tools and lessons learned by the NILM research community, to other machine-learning domains.

**Author Contributions:** Conceptualization and Methodology: L.P.; Software: M.P. and N.V.; Writing-original draft preparation: L.P. and M.P.; Writing-review and editing: L.P., M.P. and N.V.

**Funding:** This research was funded by the regional project M1420-01-0145-FEDER-000002 co-financed by "Madeira 14-20", and by the Portuguese Foundation for Science and Technology (FCT) under projects UID/EEA/50009/2019, and CEECIND/01179/2017.

**Acknowledgments:** The authors thank the anonymous reviewers for their constructive comments.

**Conflicts of Interest:** The authors declare no conflict of interest.

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
