# Peer review of "dsCleaner: A Python Library to Clean, Preprocess and Convert Non-Intrusive Load Monitoring Datasets"

_data, 2019_

Round 1

Reviewer 1 Report

This paper presents the DSCleaner, a Python library to clean, pre-process and convert time series datasets to a common file format. Two examples are used to demonstrate the effectiveness of the library. Overall, this is a good technical note. The following suggestions are for the authors' reference:

       (1) The introduction lacks a description of time series data. The NILM dataset is just one class of time series data, not all of them. Therefore, the problems of time series data preprocessing should also be explained in a broad sense.

       (2) The media format is somewhat obtrusive, and authors should first state the necessity of this format. Moreover, the example should not only mention two data sets, but also include the actual processing of multimedia data.

       (3) The library overview should be described in more detail. The reviewer has some difficulty in reading this section.

Author Response

Thank you for your review and suggestions. Please find our responses in the attached file.

Kind regards,

The authors.

Reviewer 2 Report

The paper does not really demonstrate/prove the main claim. The case studies shown are very specific examples and we do not know whether the transformed datasets are usable at all. For example, why is resampling necessary? Why did you reduce the number of files? Maybe having more number files is necessary so that less amount of data can be in use at a time (rather than reading a one big file). In general in a work like this, it would be better to first see a user analysis (from actual data set users) that surveys them regarding what to be included. The current version just demonstrates the authors' own data preprocessing methodology, not a common data preprocessing method that could be applicable for all data sets, all users, or all applications. 

Author Response

Thank you for your comments, Please see the attachment for our responses.

Kind regards,

The authors.

Round 2

Reviewer 1 Report

The paper can be accepted in present form.

Author Response

Thank you.

Reviewer 2 Report

I am quite happy with the current direction of the paper and how the authors have fixed the focus of the paper to NILM datasets. This quite improved the paper. I suggest the following additions to the current version of the paper if applicable: 

- The interoperability of the work is stressed as an important feature of the work. Is it possible to discuss this more and elaborate after describing the datasets worked on? 

- It would be nice to see what kind of properties this work provides (modality, interoperability, etc.) as a part of the functionalities listed in the Introduction section. 

- It also would be better to connect the properties/functionalities to the classes described in section 2.2. 

- Is it possible to have any quantified result other than the reduced file size metric?

- Are there any other related work to add to the paper? This would strengthen the arguments in the Introduction section. 

Author Response

Thank you for your valuable comments.

Please find our replies and in the attachment.

Round 3

Reviewer 2 Report

I would like to thank the authors for addressing my comments and improving the paper substaintially.